# Performance of deep learning to detect mastoiditis using multiple conventional radiographs of mastoid

**Kyong Joon Lee**[1], **Inseon Ryoo**[2]*, **Dongjun Choi**[1], **Leonard Sunwoo**[1], **Sung-Hye You**[3], **Hye Na Jung**[2]

**1** Department of Radiology, Seoul National University Bundang Hospital, Gyeonggi-do, Korea, **2** Department of Radiology, Korea University Guro Hospital, Korea University College of Medicine, Seoul, Korea, **3** Department of Radiology, Korea University Anam Hospital, Seoul, Korea

\* isryoo@gmail.com

**Data Availability Statement:** All relevant data are within the manuscript and its Supporting Information files.

## Abstract

### Objectives

This study aimed to compare the diagnostic performance of deep learning algorithm trained by single view (anterior-posterior (AP) or lateral view) with that trained by multiple views (both views together) in diagnosis of mastoiditis on mastoid series and compare the diagnostic performance between the algorithm and radiologists.

### Methods

Total 9,988 mastoid series (AP and lateral views) were classified as normal or abnormal (mastoiditis) based on radiographic findings. Among them 792 image sets with temporal bone CT were classified as the gold standard test set and remaining sets were randomly divided into training (n = 8,276) and validation (n = 920) sets by 9:1 for developing a deep learning algorithm. Temporal (n = 294) and geographic (n = 308) external test sets were also collected. Diagnostic performance of deep learning algorithm trained by single view was compared with that trained by multiple views. Diagnostic performance of the algorithm and two radiologists was assessed. Inter-observer agreement between the algorithm and radiologists and between two radiologists was calculated.

### Results

Area under the receiver operating characteristic curves of algorithm using multiple views (0.971, 0.978, and 0.965 for gold standard, temporal, and geographic external test sets, respectively) showed higher values than those using single view (0.964/0.953, 0.952/0.961, and 0.961/0.942 for AP view/lateral view of gold standard, temporal external, and geographic external test sets, respectively) in all test sets. The algorithm showed statistically significant higher specificity compared with radiologists (p = 0.018 and 0.012). There was substantial agreement between the algorithm and two radiologists and between two radiologists (κ = 0.79, 0.8, and 0.76).

**Funding:** IR, Grant No: 2017R1C1B5076240, National Research Foundation of Korea, URL: www.nrf.re.kr; KJL, Grant No: 13-2019-006, Seoul National University Bundang Hospital Research Fund, URL: www.snubh.org. The funders had no role in study design, data collection and analysis, decision to publish, or preparation of the manuscript.

**Competing interests:** The authors have declared that no competing interests exist.

## Conclusion

The deep learning algorithm trained by multiple views showed better performance than that trained by single view. The diagnostic performance of the algorithm for detecting mastoiditis on mastoid series was similar to or higher than that of radiologists.

## Introduction

Otomastoiditis is the second most common complication of acute otitis media (AOM) after tympanic membrane perforation [1]. Over the last several decades, the incidence of otomastoiditis as a complication of AOM has greatly decreased [1,2]. Nevertheless, improper management with antibiotics cannot prevent otomastoiditis and the incidence of otomastoiditis remains at approximately 1% [1]. Furthermore, the increasing numbers of immunocompromised patients who underwent organ transplantation surgery or received chemotherapy also increases the incidence of the occurrence of otomastoiditis. Diagnosis and early adequate treatment of otomastoiditis is very important, as the complications of otomastoiditis include sinus thrombosis and thrombophlebitis (transverse or sigmoid sinuses), encephalitis, and meningitis due to its proximity to intracranial structures [1,3].

High resolution temporal bone CT (TB CT) is the imaging modality of choice for the diagnosis of mastoiditis [4,5]. However, plain radiography of the mastoid (mastoid series) is still effective for screening mastoiditis in populations with very low prevalence such as pre-transplantation operation work up. Moreover, because the most commonly affected age group is the pediatric group, especially patients under two years old who are very sensitive to radiation exposure, simple radiography still has its role [2,6].

Because of the advancement in medical imaging techniques over the last century, there has been a tremendous increase in the amount of medical images. Simple radiographies are still the most commonly performed medical imaging until now due to both cost-effectiveness and clinical usefulness [7,8]. Therefore, simple radiographies take a large portion of radiologists' work-loads [7–9]. In addition, accurate interpretation of simple radiographs requires an extensive amount of medical and radiologic knowledge and experience, as simple radiographies are composed of complex three-dimensional anatomic information projected into two-dimensional images [8].

Many studies have explored the application of deep learning technology to interpret simple radiographs (i.e., chest posterior-anterior [PA], Waters view, and mammography mediolateral oblique [MLO] view) to solve current problems in clinical practices including explosively increased radiologists' work-loads and the intrinsic challenges of interpreting simple radiographs [10–14]. However, most studies used single-view images rather than multiple view images, unlike daily practices which usually use multiple view images in diagnosing diseases.

In this study, we developed a deep learning algorithm with a large dataset and evaluated its diagnostic performance in detecting mastoiditis with a mastoid series compared to the performance by head and neck radiologists. We also compared the diagnostic performance of the algorithm using multiple views (mastoid anterior-posterior [AP] view with lateral view) with that using single view (AP view or lateral view only).

## Materials and methods

The Institutional Review Boards of Korea University Guro Hospital approved this study and informed consent was waived considering the retrospective design and anonymized data used in this study.

## Dataset

Mastoid series for screening mastoiditis from 5,156 patients were collected from Korea University Guro Hospital (KUGH) that were taken between April 2003 and November 2018. Mastoid series were performed for screening mastoiditis not only in patients with suspected mastoiditis but also in patients planning to receive operations such as organ transplantation and cochlear implantation. In those cases, preoperative treatment of mastoiditis very important. The mastoid series consisted of an AP view and two lateral views (i.e., left and right lateral views). We excluded images that did not contain either the AP view or the bilateral views or the images that were uninterpretable due to artifacts. One hundred sixty-two patients were excluded according to these exclusion criteria. Among the remaining 4,994 patients, series of 396 patients who underwent TB CT examinations within seven days of the study date of their mastoid series were set aside as the gold standard test set. Majority of cases in the gold standard test set performed TB CT and mastoid series simultaneously. The mastoid series of the other 4,598 patients were used as the dataset for developing our deep learning algorithm. The training set and the validation set were generated by randomly dividing the dataset by 9:1.

We also collected temporally and geographically external test sets to further comprehensively verify the performance of the deep learning algorithm. The temporally external test set was collected from KUGH from December 2018 to April 2019 in 150 patients, and three patients were not included according to the exclusion criteria. The geographically external test sets were collected from Korea University Anam Hospital in 154 patients with the same time period.

We designed the deep learning algorithm to adopt an image set for one individual ear and to yield a classification result for the ear. An AP view was divided by the vertical bisector, and each half was fed into the algorithm as one individual training sample. A lateral view was directly used as an input training sample because the right and left lateral views already existed in separate images.

## Labeling

Digital Imaging and Communication in Medicine (DICOM) files of the mastoid series were downloaded from the picture archiving and communication system (PACS) and all image data were anonymized for further analyses.

Two head and neck neuroradiologists (I.R. and H.N.J., both with 12 years of experience in this field) independently labeled mastoid series of the training and validation sets, temporal external test set, and geographic external test set and the labels were determined by consensus after the two radiologists discussed. The training and validation sets (mastoid series of 4,598 patients, total image sets of 9,196 ears) and the temporal (147 patients, 294 ears) and geographic (154 patients, 308 ears) external test sets were labeled based on the radiographic findings, whereas the gold standard test sets (mastoid series of 396 patients, image sets of 792 ears) were labeled based on the results of concurrent TB CT by one reader (I.R.).

For comparison of the diagnostic accuracy of the algorithm with head and neck neuroradiologists' accuracy, two head and neck neuroradiologists with 12 years and 11 years of experience in this field (I.R. and L.S.) labeled mastoid series of the gold standard test set (images of 792 ears from 396 patients). The labeling criteria were same as the criteria used in labeling the training and validation sets.

Labeling TB CT and labeling mastoid series of gold standard test set were performed separately. Furthermore, mastoid series of gold standard test set were randomly mixed with other mastoid series in the training set and validation set when labeled.

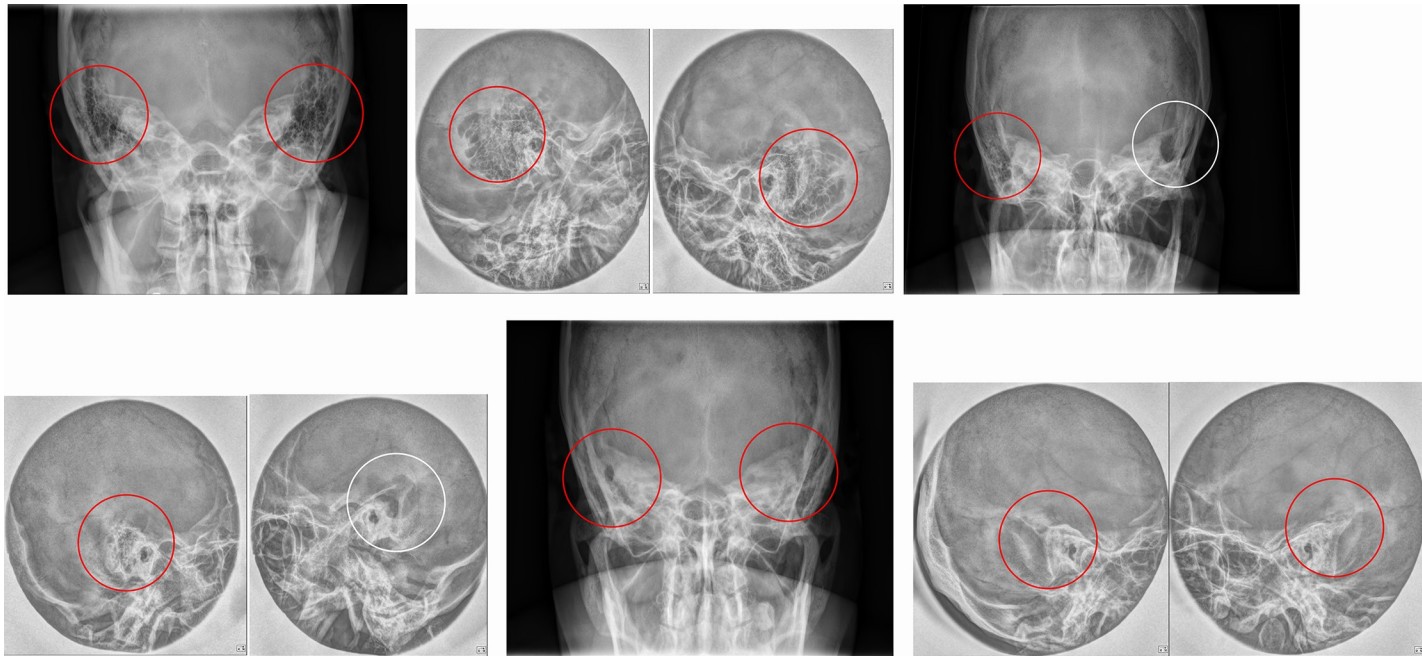

**Fig 1. Typical images of each labeling category.** (a,b) AP view (a) and lateral view (b) show bilateral, clear mastoid air cells (red circles) with honey combing pattern of category 0. (c,d) right ear (red circles) of AP view (c) and lateral view (d) shows slightly increased haziness in mastoid air cells suggesting category 1 and left ear (white circles) of both views shows bony defects with air cavities suggesting category 3. (e, f) AP view (e) and lateral view (f) show bilateral, total haziness and sclerosis of mastoid air cells (red circles) suggesting category 2.

All the images in the dataset were labeled according to the following criteria in 5 categories: category 0, normal, clear mastoid air cells on both views (Fig 1A and 1B); category 1, mild, some haziness of mastoid air cells on any of AP view and lateral view (right ear in Fig 1C and 1D); category 2, severe, total haziness and sclerosis of mastoid air cells in both the AP and lateral views (Fig 1E and 1F); category 3, mastoidectomy state (left ear in Fig 1C and 1D); and category 4, unable to be labeled due to artifacts. Data sets labeled as category 3 or 4 were excluded from the training and validation sets and external validation sets. There were too few cases of postoperative images (category 3) to include for further analyses. However, postoperative cases (category 3) were included in the gold standard test set to evaluate how the algorithm classified the data. The gold standard test sets were labeled as category 0, normal; category 1, mild, soft tissue densities in some mastoid air cells; category 2, severe, soft tissue densities in near total air cells with sclerosis; and category 3, postoperative state according to the results of TB CT.

To simplify the interpretation of the results and to address the class imbalance issue due to the lack of positive samples, the labels were dichotomized with 0 as normal (category 0) and 1 as abnormal (category 1 and 2). The postoperative state (category 3) in the gold standard test set was also set as abnormal [1].

## Deep learning algorithm

The AP view and the lateral views underwent the following preprocessing step before applying the deep learning algorithm. We cropped both ears in AP views assuming that all the images were taken at regular positions. For instance, the right ear in an AP view was cropped to have a size of 180×120 mm centered on 0.6 and 0.25 times the coordinates of the original image height and width and then the image was resized to 384×256 pixels. The left ear on the AP view was cropped in a similar way centered on the symmetrical coordinates. The right and left

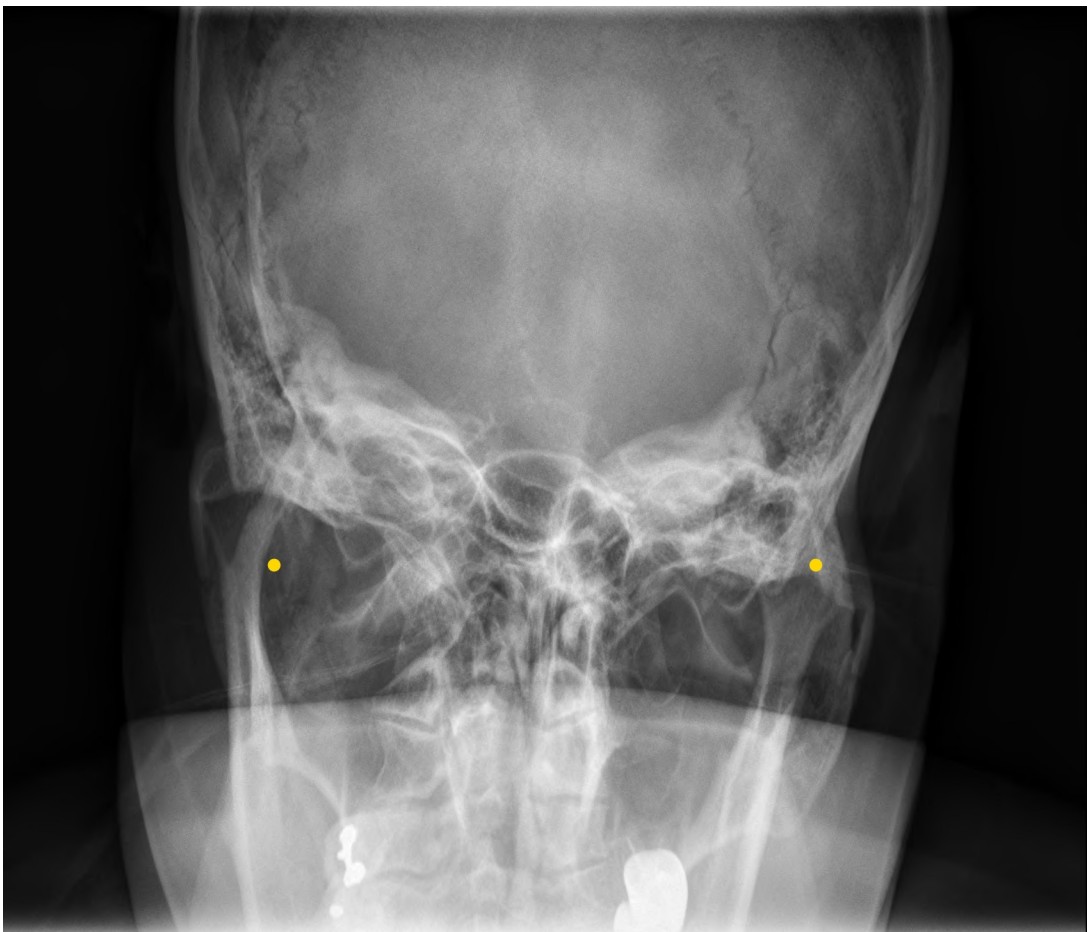

**Fig 2. Location of center points for right and left cropping in AP view.** The yellow dots represent the center points of cropping the right/left ears.

center points in the AP view are the points annotated in Fig 2. Outliers were excluded from the analyses. The lateral view was cropped to have a size of 140×140 mm at the center coordinates of the original image and was resized to 256×256 pixels. For data augmentation, horizontal and vertical shift and horizontal flipping were applied in the training set. We used the Pydicom library (Python Software Foundation; version 1.2.0) to process the images in the DICOM format.

We performed the training on CUDA/cuDNN (versions 9.0 and 7.1, respectively) and TensorFlow library (version 1.12) for graphic processing unit acceleration on a Linux operating system. OS, CPU and GPU were Ubuntu 16.04, Intel® Xeon® CPU E5-2698 v4 @ 2.20GHz 80 cores, and Tesla V100-SXM2-32GB, respectively.

We designed two neural networks: i.e., one for a single view (an AP view or a lateral view, one image at a time, Fig 3A) and the other for multiple views (an AP view and two lateral views simultaneously, Fig 3B).

The convolutional neural network (CNN) for the single view consisted of a stack of six squeeze-and-excitation ResNet (SE-ResNet) modules [15] followed by the Log-Sum-Exp pooling [16] applied to the last SE-ResNet module (Fig 3A). Mastoiditis was predicted by applying Sigmoid function to the output of Log-Sum-Exp pooling. The weights of the network were initialized by Xavier initialization [17]. The learning rate decayed every 5,000 steps at the rate of

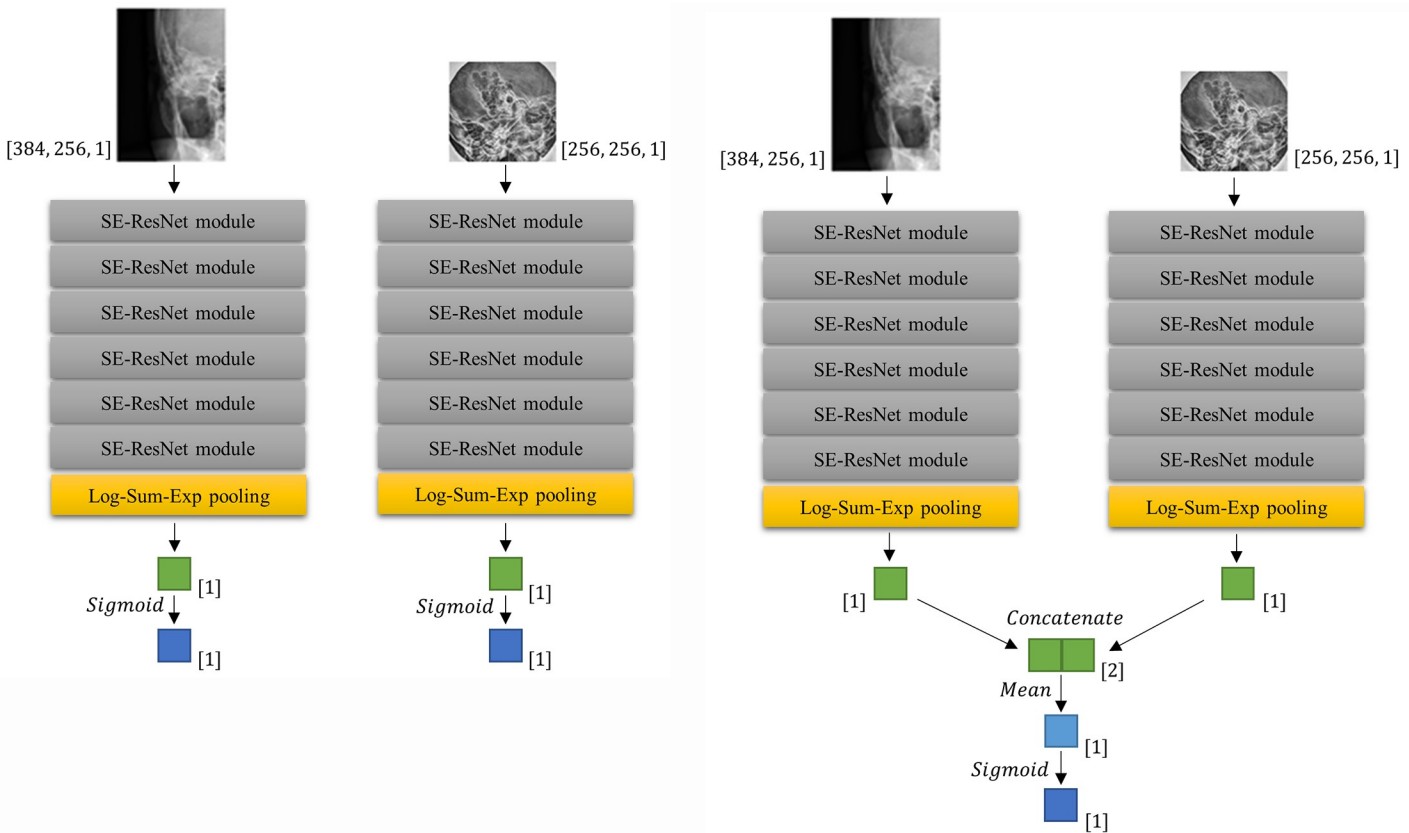

**Fig 3. Network architectures for predicting mastoiditis.** The CNNs (convolutional neural networks) for single view (a) show a process in which AP and lateral views are separately trained in CNN. The CNN for multiple views (b) shows a process in which AP and lateral views are simultaneously trained. After Log-Sum-Exp pooling, the layers were also marked with dimensions. [1] means 1×1 size vector, and [2] means 1×2 size vector.

0.94 with the initial value of 0.01. The Cross-Entropy loss was minimized by employing the RMSProp optimizer [18], L2 regularization was applied to prevent overfitting, and batch size was set to 12.

The CNN for multiple views was constructed by combining the CNN applied to each view (Fig 3B). Each CNNs for single view were combined by concatenate and average the output of Log-Sum-Exp pooling of each single view network without additional weight. Finally, Sigmoid function was applied at the averaged value to predict mastoiditis. To enhance efficacy, we started training with the weights obtained from the training of each single views for the pre-training [19]. The learning rate decayed every 1,000 steps at the rate of 0.04 with 0.005 as the initial value smaller than in CNN for single view to fine-tune already gained weights. Loss function, optimizer, regularization term, and batch size were the same as for the CNN for single view. The CNNs we implemented were uploaded to the public repository (https://github.com/djchoi1742/Mastoid_CNN).

A class activation map was generated to identify which parts of the original image were activated when the CNN recognized mastoiditis. In the CNN both single view and multiple views, the class activation mappings of each view were obtained by resizing to their input size using bilinear interpolation based on the results immediately before the Log-Sum-Exp pooling step in Fig 3A and 3B. The class activation mappings were obtained by applying rectified linear activation function (ReLU) to these results in order to confirm the region strongly predicted to have mastoiditis. Since the presence or absence of mastoiditis was calculated by Sigmoid

function as a final output, the region predicted to have mastoiditis was detected only on the class activation mapping for the image judged to have mastoiditis. Otherwise, no region was detected.

## Statistical analysis

We used DeLong's test for two correlated receiver operating characteristic curves (ROC curves) [20] to compare the diagnostic performance of the algorithm using multiple views with that using a single view. Sensitivity, specificity, and area under the receiver operating characteristic curve (AUC) were used as measures to evaluate the performance of the deep learning algorithm. We applied three cut-off points calculated from the validation set to the test datasets; i.e., the optimal cut-off point obtained by Youden's J statistic, the cut-off point at which sensitivity was 95%, and the cut-off point at which specificity was 95%. Clopper-Pearson method [21] was applied to calculate 95% confidence intervals for sensitivity and specificity, and the method of DeLong et al. [20] was used to calculate 95% confidence intervals of AUC.

McNemar's test for sensitivity and specificity was used to compare the diagnostic performance between the deep learning algorithm and the radiologist results. Cohen's $\kappa$ coefficient was used to evaluate the agreement between the results diagnosed by deep learning algorithm and the radiologist diagnosis. The level of agreement was interpreted as poor if $\kappa$ was less than 0; slight, 0 to 0.20; fair, 0.21 to 0.40; moderate, 0.41 to 0.60; substantial, 0.61 to 0.80; and almost perfect, 0.81 to 1.00 [22]. Qualitative analysis was performed by showing the CNN class activation map [23].

All statistical analyses were performed with R statistical software version 3.6.1 (The R Foundation for Statistical Computing, Vienna, Austria). A p-value less than 0.05 was considered statistically significant.

## Results

The baseline characteristics of training set, validation set, gold standard test set, temporal external test set, and geographic external test set are shown in **Table 1**. There was no statistical difference in label distribution between the temporal external test set and geographic external test set (P = 0.118). The ROC curves and DeLong's test for two correlated ROC curves comparing the diagnostic performance between the algorithm using single view and the algorithm using multiple views in each dataset are shown in **Fig 4** and **Table 2**, respectively. In comparison of the diagnostic performance of deep learning algorithm using multiple views (AP view and lateral view) with that using a single view (AP view or lateral view only), AUCs from the multiple views showed statistically significant higher values than AUCs using a single view (AP view or lateral view only) in the validation set and all test sets (gold standard test set, temporally external test set, and geographically external test set), except for AUC using a single AP view in the geographic external test set. Even the AUC using AP view in the geographic external test set also showed a lower value than AUC using multiple views; however, statistical significance was not shown (P = 0.246).

The sensitivity and specificity calculated based on the optimal cut-off point determined by Youden's J statistic, the cut-off point for an expected sensitivity of 95%, and the cut-off point for an expected specificity of 95% described above to evaluate the diagnostic accuracy of the deep learning algorithm and radiologist results are summarized in **Table 3**. Usually, cut-off point is derived from the validation set assuming that we do not know the exact distribution of the test set [24]. This data was based on labels diagnosed with TB CT. With the optimal cut-off point, the sensitivity and specificity of the gold standard test set diagnosed by the deep learning algorithm were 96.4% (423/439, 95% confidence interval, 94.1% - 97.9%) and 74.5% (263/353,

**Table 1. Baseline characteristics of all data sets.**

| Characteristic | | | Training set (n = 8278) | Validation set (n = 918) | Test sets | | |
| --- | --- | --- | --- | --- | --- | --- | --- |
| | | | | | Gold standard test set (n = 792) | Temporal external test set (n = 294) | Geographic external test set (n = 308) |
| Number of patients | | | 4139 | 459 | 396 | 147 | 154 |
| Age | | | | | | | |
| | <20 years | | 322 | 35 | 40 | 4 | 4 |
| | 20~29 years | | 262 | 32 | 16 | 4 | 7 |
| | 30~39 years | | 444 | 68 | 47 | 10 | 15 |
| | 40~49 years | | 912 | 97 | 85 | 21 | 28 |
| | 50~59 years | | 1131 | 114 | 119 | 60 | 59 |
| | 60~69 years | | 774 | 80 | 55 | 37 | 26 |
| | 70~79 years | | 258 | 30 | 27 | 8 | 14 |
| | ≥80 years | | 36 | 3 | 7 | 3 | 1 |
| Sex | | | | | | | |
| | Female | | 2353 | 247 | 225 | 70 | 69 |
| | Male | | 1786 | 212 | 171 | 77 | 85 |
| Label (based on conventional radiography) | | | | | | | |
| | 0, Normal | | 3155 | 349 | 261 | 159 | 175 |
| | 1, Abnormal | | 5123 | 569 | 531 | 135 | 133 |
| | | Mild | 1806 | 200 | 129 | 56 | 44 |
| | | Severe | 3317 | 369 | 402 | 76 | 89 |
| | | Postop | - | - | - | 3 | - |
| Label (based on CT) | | | | | | | |
| | 0, Normal | | - | - | 353 | - | - |
| | 1, Abnormal | | - | - | 439 | - | - |
| | | Mild | - | - | 157 | - | - |
| | | Severe | - | - | 258 | - | - |
| | | Postop | - | - | 24 | - | - |
| κ coefficient between two reviewers | | | 0.78 | 0.77 | 0.76 | 0.79 | 0.79 |

95% confidence interval, 69.6% - 79.0%), respectively. The sensitivity of the deep learning algorithm was not significantly different from those of the radiologists (p-value = 0.752 and 1.000). However, the specificity diagnosed by the deep learning algorithm was significantly higher than those of the radiologists (p-value = 0.018 and 0.012, respectively). In addition, confusion matrices between the prediction of the deep learning algorithm and radiologists' predictions based on the reference standard are shown in **Fig 5**. Gold standard labels were divided into categories 1, 2, and 3 to check how the deep learning algorithm and radiologists predicted normal or abnormal and to see the detailed results of each label. The postoperative category (category 3) was not included in training; however, this was considered abnormal in the gold standard test set. Severe (category 2) labels were predicted as abnormal except for two cases, and mild (category 1) labels were predicted as abnormal at about 90%; all postoperative data were predicted as abnormal.

There was substantial agreement between the radiologists and deep learning algorithm (κ coefficient between radiologist 1 and deep learning algorithm: 0.79, κ coefficient between radiologist 2 and deep learning algorithm, 0.8) in the gold standard test set. In addition, the κ coefficient between radiologist 1 and radiologist 2 was 0.76 in the same test set.

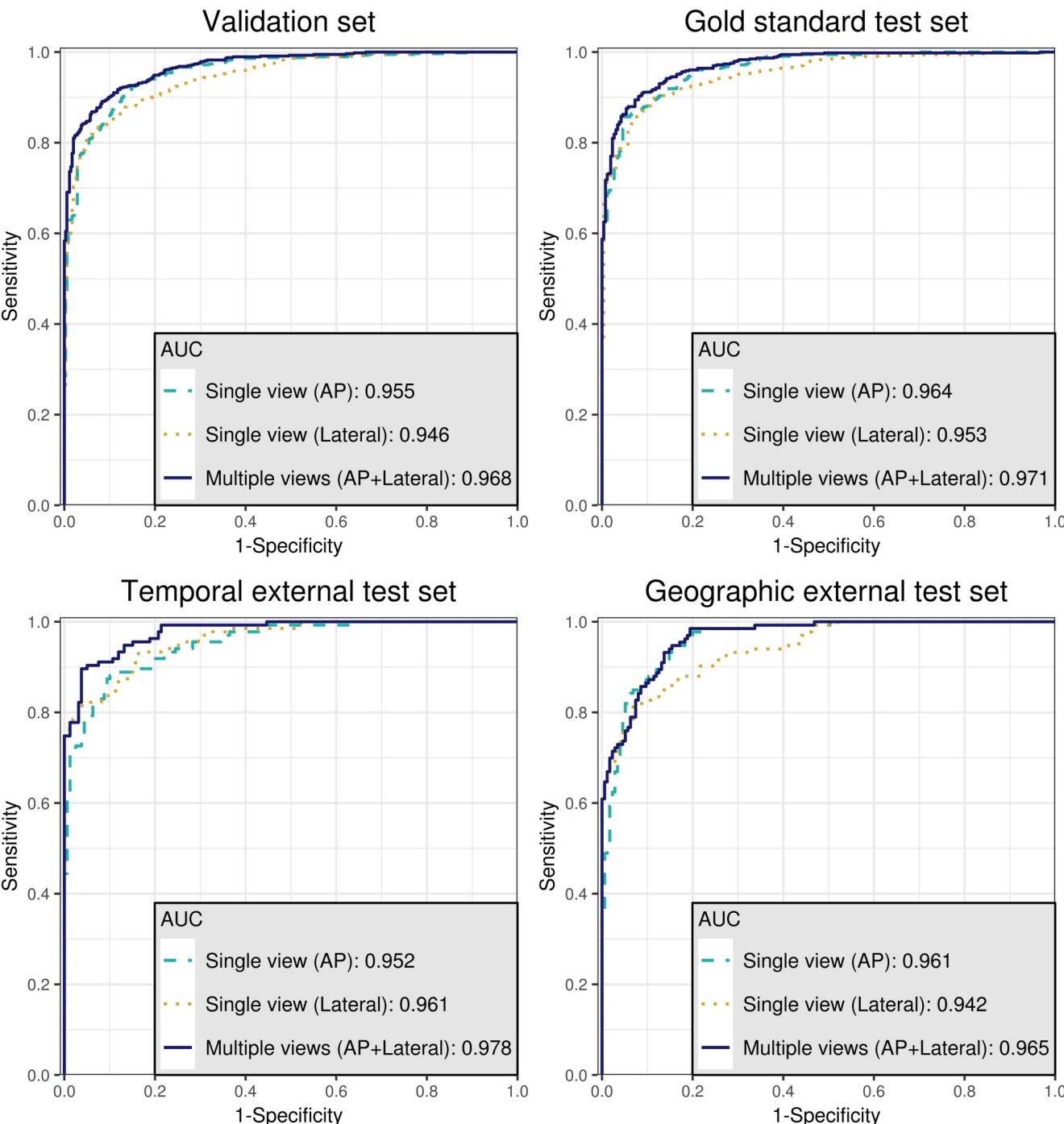

**Fig 4. The receiver operating characteristic curves (ROC curves) for validation set and three test sets.** The area under the ROC curves (AUCs) using the multiple views show statistically significant higher values than AUCs using a single view (AP view or lateral view only) in the validation set and all test sets.

The AUC, sensitivity, and specificity of the test sets were calculated based on the labels diagnosed based on both AP view and lateral view. The AUC of the deep learning algorithm for gold standard test set, temporal external test set, and geographic external test set was 0.97 (95% confidence interval, 0.96–0.98), 0.98 (95% confidence interval, 0.97–0.99), and 0.97 (95% confidence interval, 0.95–0.98), respectively. The sensitivity and specificity of the deep learning

**Table 2. Comparison of the diagnostic performance between the algorithm using single view and the algorithm using multiple views in each data set based on labels by conventional radiography.**

| Dataset | Comparison | AUC (single view) | AUC (multiple views) | $P^*$ |
|---|---|---|---|---|
| Validation set | Single view (AP) vs Multiple views | 0.955 (0.943–0.968) | 0.968 (0.959–0.977) | <0.001* |
| | Single view (Lateral) vs Multiple views | 0.946 (0.932–0.959) | | <0.001* |
| Gold standard test set | Single view (AP) vs Multiple views | 0.964 (0.953–0.975) | 0.971 (0.962–0.981) | 0.017* |
| | Single view (Lateral) vs Multiple views | 0.953 (0.940–0.966) | | <0.001* |
| Temporal external test set | Single view (AP) vs Multiple views | 0.952 (0.931–0.974) | 0.978 (0.965–0.990) | 0.002* |
| | Single view (Lateral) vs Multiple views | 0.961 (0.942–0.980) | | 0.004* |
| Geographic external test set | Single view (AP) vs Multiple views | 0.961 (0.942–0.980) | 0.965 (0.948–0.981) | 0.246 |
| | Single view (Lateral) vs Multiple views | 0.942 (0.918–0.966) | | 0.003* |

Data is shown to three decimal places, with the 95% confidence interval in parentheses.

AUC: Area under the receiver operating characteristic (ROC) curves.

$P^*$: P-value of one-side DeLong's test for two correlated ROC curves (Alternative hypothesis: AUC of multiple views was greater than AUC of single view).

*:<0.05 was significant.

algorithm of the three test sets are shown in **Table 4**. The sensitivity and specificity of the gold standard test set were 91.3% (95% confidence interval, 88.6–93.6%) and 89.3% (95% confidence interval, 84.9–92.8%), respectively. The sensitivity and specificity of the temporal external test set were 91.1% (95% confidence interval, 85.0–95.3%) and 90.6% (95% confidence interval, 84.9–94.6%), respectively, and those of the geographic external test set were 85.7% (95% confidence interval, 78.6–91.2%) and 90.3% (95% confidence interval, 84.9–94.2%), respectively. Under the optimal cut-off of the validation set, the sensitivity of the geographic test set was somewhat lower than that of the other two test sets. Confusion matrices between the conventional radiography-based label and predictions of the deep learning algorithm are shown in **Fig 6**. In all three test sets, the proportion of cases incorrectly diagnosed as normal in the mild labeled group was larger than that in the severe labeled group.

Class activation mappings of the sample images are shown in **Fig 7** (true positive (a), true negative (b), false positive (c), false negative (d), and postoperative state (e) examples). These class activation mappings are obtained from the CNN for multiple view. If deep learning algorithm determined a case to be normal, neither AP view nor lateral view detected a specific region of the image (Fig 7B and 7D). In contrast, a case was determined as mastoiditis, lesion-related regions were detected in at least one of AP and lateral views (Fig 7A, 7C and 7E).

**Table 3. Comparison of diagnostic performance for gold standard test set between the deep learning algorithm (using multiple views) and radiologists based on the labels by standard reference (temporal bone CT).**

| Reader | | Sensitivity | $P^{se}$ | Specificity | $P^{sp}$ |
|---|---|---|---|---|---|
| Deep learning algorithm | Optimal cutoff | 96.4% (423/439, 94.1–97.9%) | | 74.5% (263/353, 69.6–79.0%) | |
| | Cutoff for 95% sensitivity | 98.6% (433/439, 97.0–99.5%) | | 58.9% (208/353, 53.6–64.1%) | |
| | Cutoff for 95% specificity | 95.7% (420/439, 93.3–97.4%) | | 79.3% (280/353, 74.7–83.4%) | |
| Radiologist | Radiologist 1 | 95.9% (421/439, 93.6–97.6%) | 0.752 | 68.8% (243/353, 63.7–73.6%) | 0.018* |
| | Radiologist 2 | 96.1% (422/439, 93.9–97.7%) | 1.000 | 68.6% (242/353, 63.4–73.4%) | 0.012* |

Data are percentages and nominator/denominator, and 95% confidence interval in the parentheses.

$P^{se}, P^{sp}$: P values for comparing sensitivities/specificities between the deep learning algorithm based on optimal cutoff and the radiologists were determined by using McNemar's test.

*:<0.05 was significant.

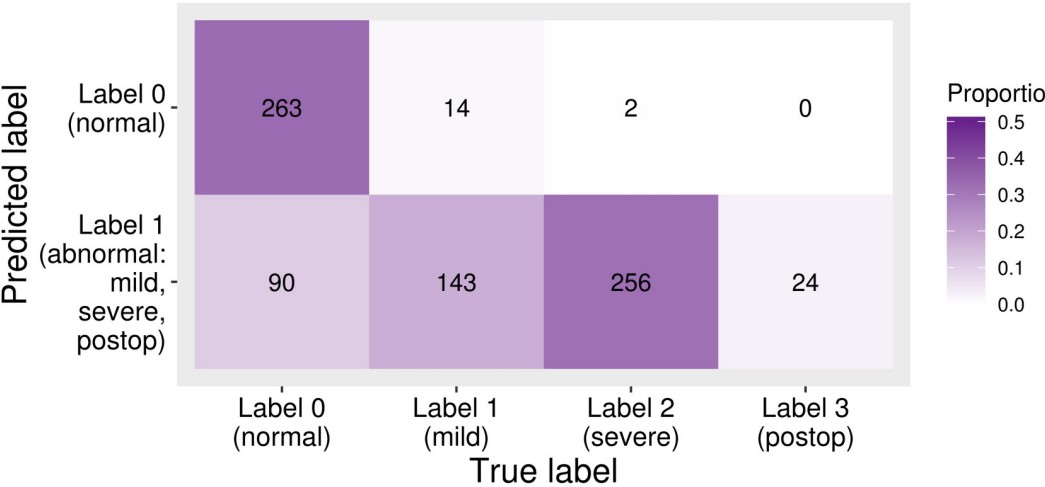

Deep learning algorithm: Gold standard test set

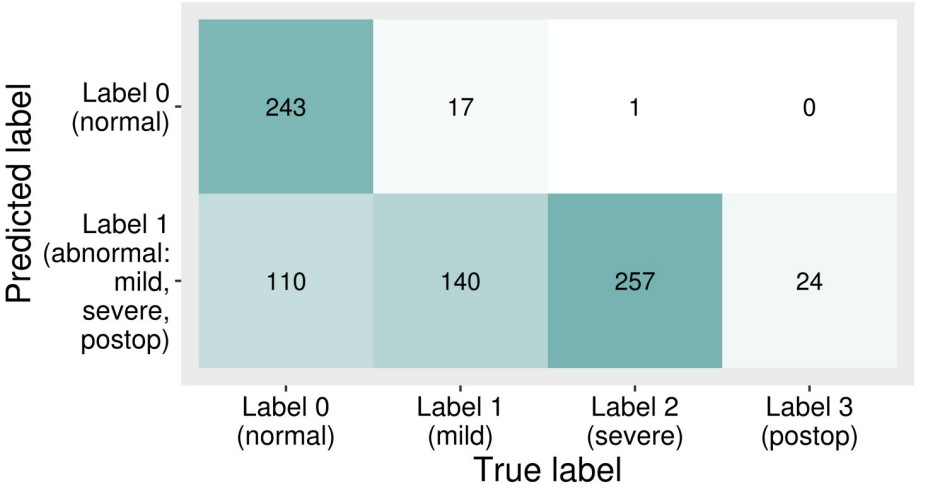

Radiologist 1: Gold standard test set

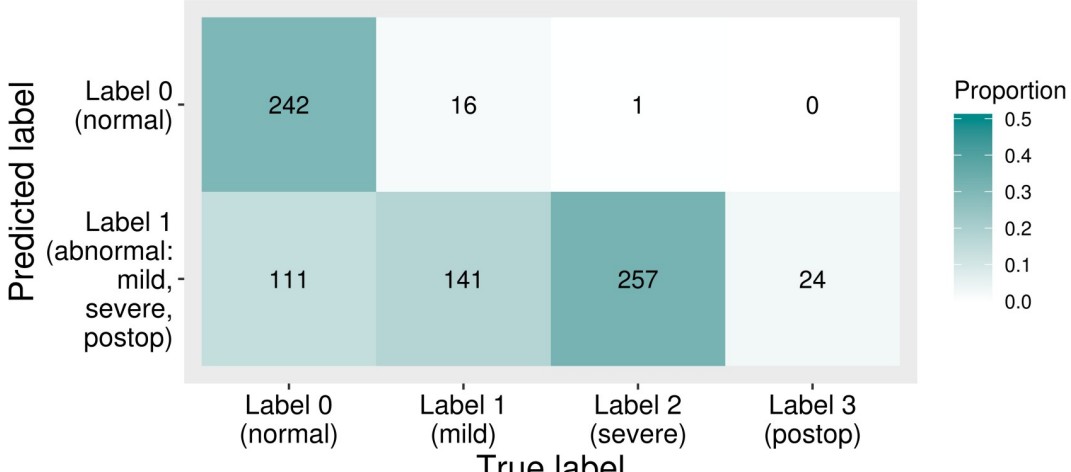

Radiologist 2: Gold standard test set

**Fig 5. Confusion matrices between predicted labels and temporal bone CT based gold-standard labels.** Predicted labels are normal (label 0) or abnormal (label 1) since the deep learning algorithm was trained based on the dichotomized data (e.g., normal or abnormal).

Although not considered in training, how the deep leaning algorithm determines both views that were labeled as postoperative state based on reference standard was also analyzed and the algorithm determined all the postoperative cases as abnormal (**Fig 7E**). In both views, abnormal regions in images were at the mastoid air cells.

## Discussion

Over the last couple of years, great advancement and progression of deep learning technologies have become integrated into not only medical fields but also all other industrial fields [25–28]. Radiology is one of the most promising fields with respect to applications for new deep learning technologies, and many previous studies have suggested vast possibilities for new directions in this field [29–33].

The amount of medical images continues to increase explosively, and the average radiologist reads more than 100 simple radiograph examinations per day in the United States [7,8]. Application of deep learning technologies to radiology can be clinically very beneficial to radiologists and can be a new academic field in radiology.

In the present study, the deep learning algorithm trained by multiple views (mastoid AP view with lateral view) of the lesion showed better performance than that trained by single view (mastoid AP view or lateral view). This is very meaningful in the interpretation of medical images, since radiologists usually use multiple images rather than a single image in clinical practice for diagnosing diseases. For example, radiologists frequently use chest PA with a lateral view for evaluation of lung diseases, Waters view with Caldwell view and lateral view for evaluation of paranasal sinusitis, and MLO view with craniocaudal (CC) view for breast cancer screening using mammography. This is not limited to simple radiographs. For advanced imaging modalities, multiphase images of a lesion are used in CT scans and even multiphase with multiple sequences of a lesion are used in MR imaging.

In this study, the deep learning algorithm could diagnose mastoiditis with accuracy similar to or higher than that of head and neck radiologists. The sensitivity (96.4%) and specificity (74.5%) of the deep learning algorithm were higher than those of head and neck neuroradiologists (sensitivities, 95.9% and 96.1%; specificities, 68.8% and 68.6%) in diagnosing mastoiditis using TB CT as the standard reference. In terms of specificity, there was a statistically significant difference.

**Table 4. Diagnostic performance of deep learning algorithm in all test sets based on labels by conventional radiography.**

| Diagnostic performance | | Gold standard test set | Temporal external test set | Geographic external test set |
|---|---|---|---|---|
| AUC | | 0.971 (0.962–0.981) | 0.978 (0.965–0.990) | 0.965 (0.948–0.981) |
| Optimal cutoff | Sensitivity | 91.3% (485/531, 88.6–93.6%) | 91.1% (123/135, 85.0–95.3%) | 85.7% (114/133, 78.6–91.2%) |
| | Specificity | 89.3% (233/261, 84.9–92.8%) | 90.6% (144/159, 84.9–94.6%) | 90.3% (158/175, 84.9–94.2%) |
| Cutoff for 95% sensitivity | Sensitivity | 96.8% (514/531, 94.9–98.1%) | 97.8% (132/135, 93.6–99.5%) | 97.0% (129/133, 92.5–99.2%) |
| | Specificity | 75.5% (197/261, 69.8–80.6%) | 79.2% (126/159, 72.1–85.3%) | 80.6% (141/175, 73.9–86.2%) |
| Cutoff for 95% specificity | Sensitivity | 89.3% (474/531, 86.3–91.8%) | 90.4% (122/135, 84.1–94.8%) | 85.0% (113/133, 77.7–90.6%) |
| | Specificity | 92.7% (242/261, 88.9–95.6%) | 93.7% (149/159, 88.7–96.9%) | 91.4% (160/175, 86.3–95.1%) |

Data are percentages and nominator/denominator and/or 95% confidence interval in the parentheses.

AUC: Area under the receiver operating characteristic curve.

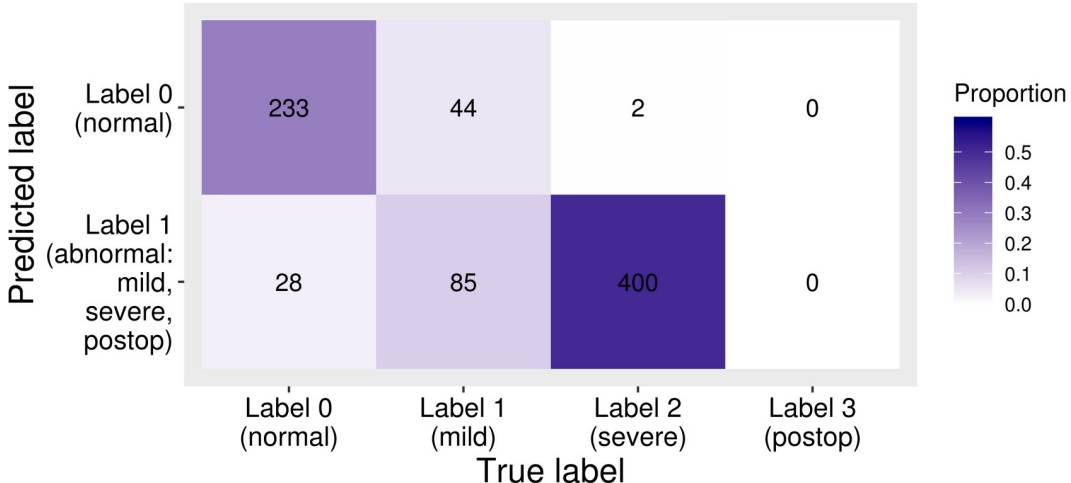

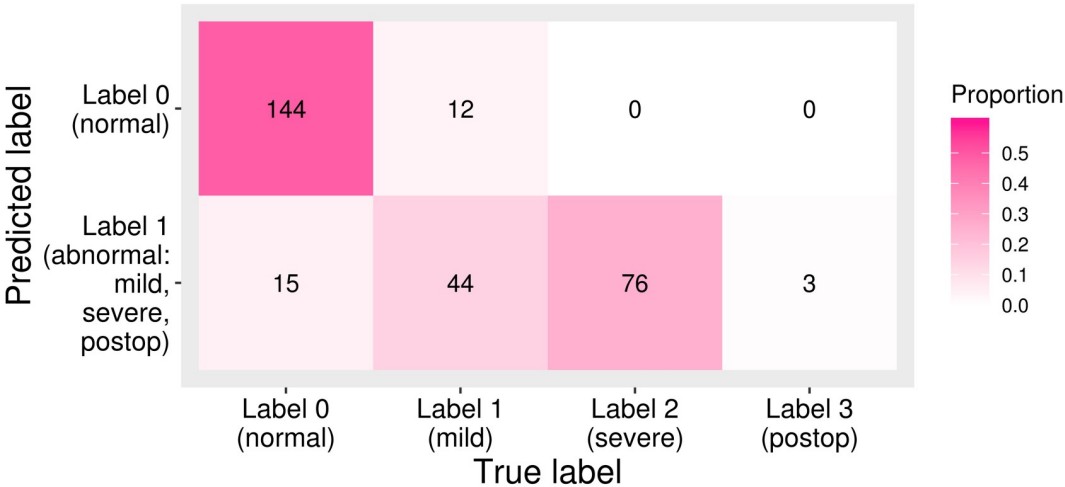

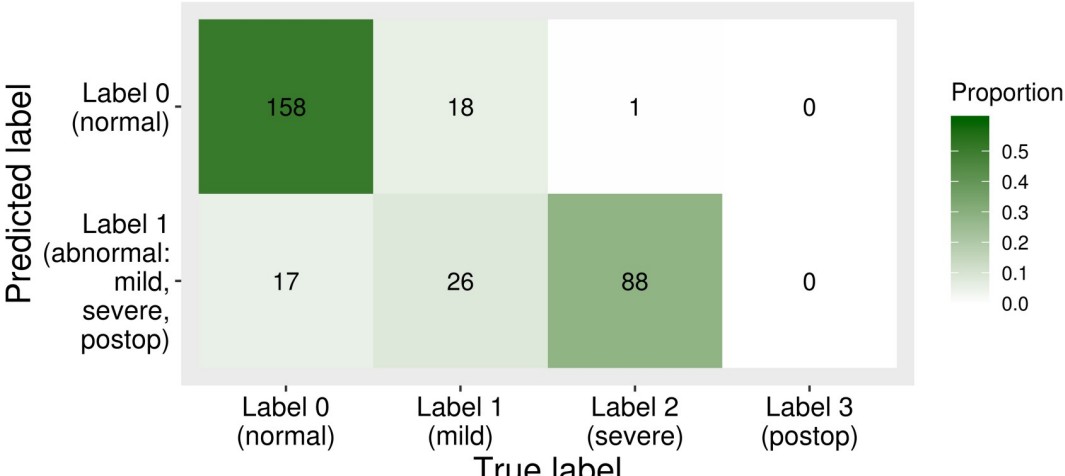

**Fig 6. Confusion matrices between the predicted labels of a deep learning algorithm and labels based on conventional radiography.** In all test sets, the proportion of the incorrectly diagnosed cases was larger in mild labeled group (category 1) than in severe labeled group (category 2).

Even though the deep learning algorithm cannot directly replace radiologists in diagnosing diseases with images, radiologists' work-loads can be reduced by the development of deep learning algorithms with very high sensitivity. In this case, algorithms would interpret a large portion of images and radiologists would check only positive or equivocal cases. This workflow would be especially helpful for areas with few radiologists or locations where access to radiologists is cost prohibitive [8,12]. Since the simple radiographs still take major part of

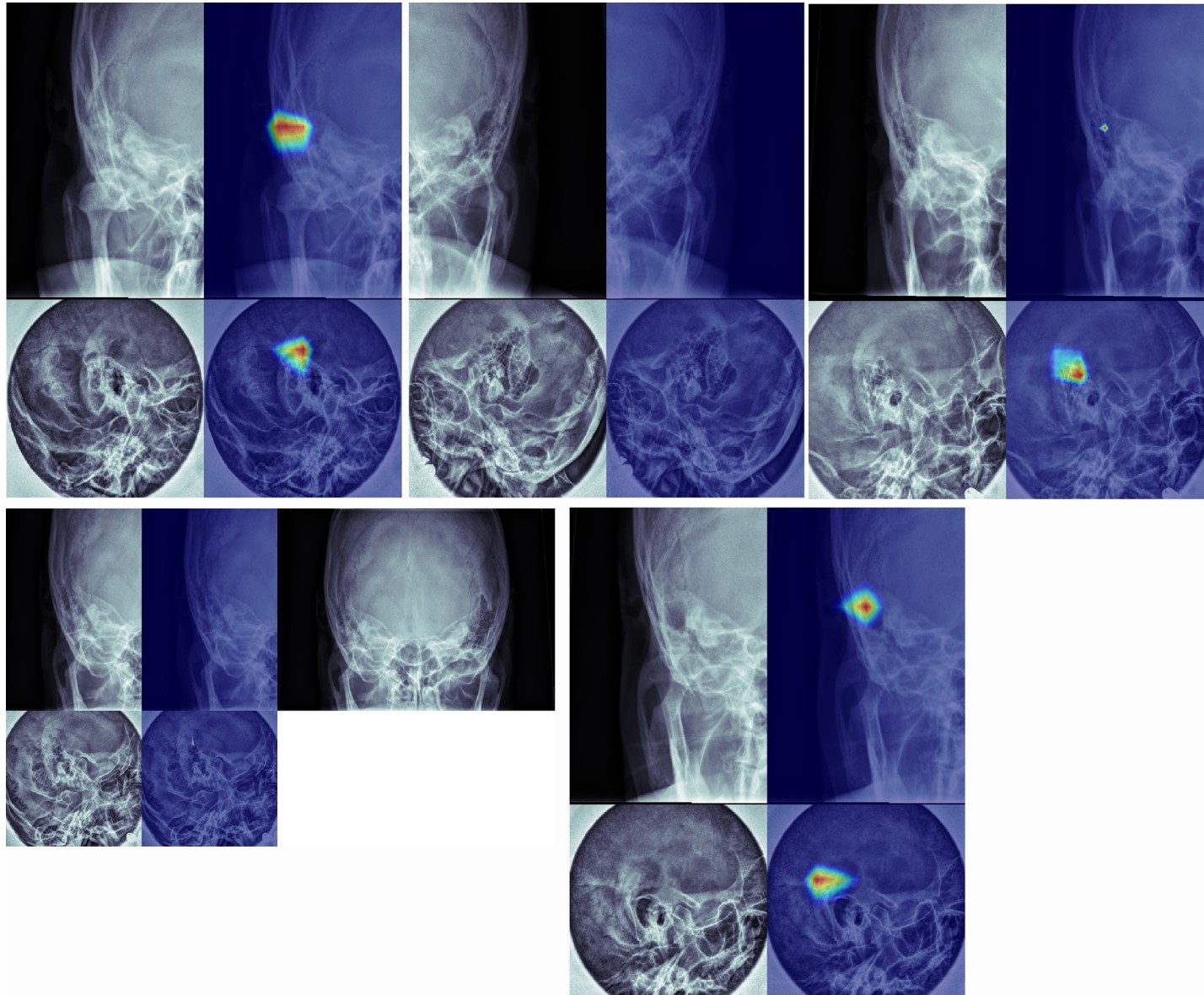

**Fig 7. Class activation mappings of true positive (a), true negative (b), false positive (c), false negative (d), and postoperative state (e) examples.** (a) Lesion related regions in mastoid air cells are detected on both AP view and lateral view. (b) No specific region is detected in either AP view or lateral view. (c) A false lesion related region is detected on lateral view. (d) Equivocal haziness is suspected on both views. The algorithm diagnosed this case as normal. An AP view with bilateral sides (right upper) shows marked asymmetry suggesting abnormality in right side. (e) Both views show lesion related regions.

radiologists' work-loads these days, deep learnings using simple radiographs are especially meaningful.

In this study, we did not simply want to evaluate the ability of the deep learning algorithm to diagnose mastoiditis with mastoid series. Unlike anatomic locations that have been studied during the last several years, including breast, paranasal sinuses, and even lung, mastoid air cells show considerable variations in terms of pneumatization between individuals and they change greatly during age [34–37]. The mastoid lateral view is also a summation of multiple complex anatomic structures such as mastoid air cells, temporo-mandibular joints, complex skull base, and even auricles. Even though there were a great deal of variations in mastoid air cells and complex anatomic structures around mastoid air cells, in this study, the class activation maps consistently showed the exact location of diseased mastoid air cells. This showed the possibility of deep learning algorithms for use in the interpretation of medical images that usually have huge anatomic diversities and variations, as long as the algorithms are trained with large enough datasets.

This study has several limitations. First, the training set and validation set had no reference standards (TB CT), and only the gold standard test set had TB CT data as a reference standard. However, the diagnostic performance of the deep learning algorithm using the gold standard test set showed better results than that using the validation set. The diagnostic performance using the external test sets also showed similar results. Second, if the deep learning algorithm found regions related to mastoiditis in only one of the two views, the algorithm often misdiagnosed a normal case as having mastoiditis and vice versa. This was similar to the actual image interpretation processes in which both views were read at the same time by head and neck radiologists. Third, radiologists used mastoid AP views with both sides of mastoid air cells simultaneously in interpreting images as used in clinical practices. In contrast, the deep learning algorithm used cropped images of the unilateral mastoid. However, this means that the deep learning algorithm was at a marked disadvantage compared with radiologists. Because there is no significant anatomic variation between bilateral mastoid air cells in one person, while there are huge variations between individuals [35], assessing symmetry of bilateral mastoid air cells in one person is very useful in the diagnosis of mastoiditis. Despite this issue, the diagnostic performance of the deep learning algorithm was similar to or higher than that of radiologists. In addition, the assessment of symmetry in human bodies on radiologic studies is a frequently used method in imaging diagnosis of diseases. Therefore, we are now developing deep learning algorithms to evaluate the symmetry of anatomies in radiologic images.

## Conclusion

This study showed that deep learning algorithm trained by multiple views of the lesion showed better performance than that trained by single view. Despite considerable anatomic variations of mastoid air cells between individuals and summation of complex anatomic structures in mastoid series, deep learning algorithm depicted the exact location of diseased mastoid air cells and showed a similar or higher performance, as compared with head and neck radiologists. Based on this result, deep learning algorithms might be applied to the interpretation of medical images that usually have huge anatomic diversities and variations, as long as trained by large enough datasets.

## Supporting information

**S1 File.**
(ZIP)

**S2 File.**
(ZIP)

**S3 File.**
(ZIP)

## Author Contributions

**Conceptualization:** Inseon Ryoo.

**Data curation:** Inseon Ryoo, Dongjun Choi, Sung-Hye You, Hye Na Jung.

**Formal analysis:** Kyong Joon Lee, Dongjun Choi, Leonard Sunwoo.

**Funding acquisition:** Inseon Ryoo.

**Investigation:** Kyong Joon Lee, Inseon Ryoo.

**Methodology:** Kyong Joon Lee, Inseon Ryoo, Dongjun Choi, Leonard Sunwoo.

**Project administration:** Inseon Ryoo.

**Resources:** Kyong Joon Lee, Inseon Ryoo.

**Software:** Kyong Joon Lee, Dongjun Choi.

**Supervision:** Kyong Joon Lee, Inseon Ryoo.

**Validation:** Kyong Joon Lee, Inseon Ryoo, Dongjun Choi, Leonard Sunwoo, Sung-Hye You, Hye Na Jung.

**Visualization:** Dongjun Choi.

**Writing – original draft:** Inseon Ryoo, Dongjun Choi.

**Writing – review & editing:** Kyong Joon Lee, Inseon Ryoo.

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
