## [Decision Letter · Decision Letter 0]

18 Jun 2020

PONE-D-20-11571

Deep learning in diagnosis of mastoiditis using multiple mastoid views

PLOS ONE

Dear Dr. Ryoo,

Thank you for submitting your manuscript to PLOS ONE. After careful consideration, we feel that it has merit but does not fully meet PLOS ONE’s publication criteria as it currently stands. Therefore, we invite you to submit a revised version of the manuscript that addresses the points raised during the review process.

We look forward to receiving your revised manuscript.

Kind regards,

Yuchen Qiu, Ph.D.

Academic Editor

PLOS ONE

Journal Requirements:

Reviewers' comments:

Reviewer's Responses to Questions

**Comments to the Author**

1. Is the manuscript technically sound, and do the data support the conclusions?

Reviewer #1: Yes

Reviewer #2: Yes

Reviewer #3: Partly

2. Has the statistical analysis been performed appropriately and rigorously? 

Reviewer #1: Yes

Reviewer #2: Yes

Reviewer #3: Yes

3. Have the authors made all data underlying the findings in their manuscript fully available?

Reviewer #1: Yes

Reviewer #2: Yes

Reviewer #3: Yes

4. Is the manuscript presented in an intelligible fashion and written in standard English?

Reviewer #1: Yes

Reviewer #2: No

Reviewer #3: Yes

5. Review Comments to the Author

Reviewer #1: This article compared the diagnostic performance of mastoiditis with Squeeze-and-Excitation Networks(SENet) trained by single view radiographic images with that trained by multiple view images and with radiologists' rating. The reference standards are the two radiologists' rating on plain radiography. Because the specificity of diagnosing mastoiditis is low on the plain radiography, the authors also provide the testing results based on radiologists' rating from CT images. The algorithms trained on 4139 patients and tested on three testing sets and showed the SENet trained on by multiple views images achieved better performance than that trained by single view images. Compare with reference standard from CT, the SENet showed similar sensitivity with radiologists and higher specificity than radiologists.

However, the authors need to find similar literature as benchmarks to compare the performance.

Authors write all data are fully available without restriction. How about the image data and labels?

The title is too general.

Line 3 add anteroposterior (AP)

Line 9 add case numbers for training and validation sets.

Line 15 add AUC values

Rephase the sentence on line 22.

Line 43 This also contributes to a tremendous increase in radiologists’ work

loads. What is this refer to?

Line 66 What is the inclusion criteria in detail? Do somehow mastoiditis need a screening? Why the mastoid series didn't include any other diseases?

What are the criteria for those patients underwent TB CT after radiography? Do you think those patients' plain radiographic images are different from the rest?

What if the two radiologists labeled different?

Show kappa coefficients between two radiologists for each dataset.

Why I.R. and L.S. instead of H.N.J. label the gold standard test set again or for CT instead of radiographic? How to get the final consensus?

It seems I.R. read both CT and plain radiography.

In the gold standard testing set, you made two types of labels, plain radiographic one and CT one. Please express clearly. And clarify it for each AUC result of gold standard testing set.

Line 102 Show a typical case for each category.

LIne 122 Why did 120*180 mm resize to 384*256 pixels? How do you determine the center point? Is there any outliers?

Line 126 Do horizontal and vertical shift on the original image or on the cropped image with zero paddings?

Line 130 add CPU GPU types and memories.

Draw the network architecture of single and multiple views, especially showing the difference of inputs.

For single view, learning rate decayed every 5000 steps at the rate of 0.94 with the initial value of 0.01. While those parameters change to 1000, 0.04, 0.005. Please explain the reason, or add new experiments to evaluate the parameter effects. The batch size is normal to set as 32,64,128 etc instead of 12.

The pediatric group is the most common. Age with mean and sd in table 1 are not good metrics. Maybe use percentage for each age group.

Clarity what the Deep learning algorithm is in table 3, multi-views or single-view?

The authors use the cut-off point at which sensitivity was 95%, and the cut-off point at which specificity was 95%. However, in the table 3 and 4. the metrics are not 95%.

Redesign table 4. It is hard to read. Maybe put sensitivity and specificity together for each point.

Which dataset is the images of Fig4 from? I think using gold-standard testing set is better due to the CT labels.

Line 372 typo "rained".

The authors did not show location results for "deep learning algorithm depicted the exact location of diseased mastoid air cells". Some selected images in Fig4 are not enough.

Line 375 rephase similar to superior... to

Reviewer #2: The authors had presented a study to compare the diagnostic performance of deep learning algorithm trained by single view or multiple views. They evaluate the performance of the algorithms trained by the two strategies and also compared those trained algorithms with expert manual diagnostic performance. The conclusion of the study is that the deep learning algorithm trained by multiple views perform better than algorithms trained by single view. Also the algorithms trained by multiple views can achieve similar (or even better) performance than the radiologists.

The conclusion drawn by the authors are meaningful, but the first part of the conclusion is predictable without the study. As stated by the authors, in practical, manual procedure would use multi-view instead of single view. Meanwhile, the accuracy of algorithms trained by using multi-view would be expected to outperform algorithms trained by using one single view. According to the ROC curves shown by the authors, even though the performances are significantly different according to statistical analysis , the accuracy numbers are not that different.

The authors have done thorough statistical analysis to support its claims. However, I believe the authors should clearly discuss the innovation or significance of this study. Currently, the paper gave out a signal that it proved an well-expected conclusion and there is limited to none innovation in the methodologies. This is a bit difficult to justify the significance of the work.

The paper is not well organized. Repetitive contents show up a lot. For instance, page 6, line 94-101 are repetitive. The paragraphs before Conclusion section are also poorly organized. The authors should re-organize the paper.

One minor question: In page 8, why are the learning rate of the two CNN with very different decay rate (0.94 vs 0.04) and initial value (0.01 vs 0.005)?

Reviewer #3: This manuscript explores mastoiditis classification with multiple view and single view and the comparison with radiologists. The manuscript is easy to understand and well-written. However, several limitations and points for further clarifications are listed below:

1. Why using the patients w/ TB CT as the gold standard test set? Is there a diagnostic accuracy difference compare to multiple view? If yes, what’s the accuracy difference?

2. The gold standard test set labeling is not clearly described in Page 6, line 94-101. Only until I read the result section, I start to realize how the labeling was conducted for the gold standard test set. I was confused by line 94 and line 100, as there are two types of labeling descriptions. Maybe the authors state ahead of that the gold standard test set was labeled twice, one was based on the concurrent TB CT by I.R. and H.N.J., the other time was based on mastoid series like in the training/validation set by I.R. and L.S.. Same for describing the labeling criteria for gold standard test set in Line 102-113, it is confusing that at the beginning saying “all the image in the dataset were labeled according…” and later on saying that “The gold standard test sets were labeled as … according to the results of TB CT.”

3. How to control if the two neuroradiologists have different opinions on the same patient, and what if the labeling is different based on TB CT and mastoid series for the gold standard test set? Which one should be used as the final classification label?

4. The deep learning method in Page 8 is not clearly described. How were the CNNs combined with multiple views? A structure figure is suggested for better illustration. Is that the CNN model is trained for each view respectively first, and further to average the last SE-ResNet module’s Log-Sum-Exp pooling values for all individual views to build the multiple view model? Is there a finetuning for the multiple view model? If yes, how did it conducted? If due to page/word count limitations, please include the details in a supplementary file.

5. In Table 1. the authors should give the full description of the abbreviations as notes. What does “CR” stands for? Please use a dash “-” to indicate the content is not available. The labels are different based on different imaging (CR and TB CT). Which is the final label of the gold standard test set, based on CR or CT? (refer to Question #3).

6. The notations of Figure 4 are not clear, I’m assuming the left side is the input image, and the right side is the outputs based on the attentions. It will be much clear if the authors can circle/point out where the lesions are in true positive (a), false negative (d), and postoperative state (e).

7. It’s suggested to provide the confusion matrix like in Fig.2 but for based on the mastoid series.

8. A normal/abnormal case is based on an individual patient or an individual ear? Is diagnostic accuracy calculated as ear-based or patient-based? If one patient has both ears as otomastoiditis, how can the authors determine the classification accuracy if the results show one ear is positive and another ear is negative?

6. PLOS authors have the option to publish the peer review history of their article (what does this mean?). If published, this will include your full peer review and any attached files.

Reviewer #1: No

Reviewer #2: No

Reviewer #3: No

---

## [Author Response · Author response to Decision Letter 0]

24 Jul 2020

We appreciate your elaborate reviews for our manuscript. We did our best to answer your comments.

Reviewer #1: This article compared the diagnostic performance of mastoiditis with Squeeze-and-Excitation Networks(SENet) trained by single view radiographic images with that trained by multiple view images and with radiologists' rating. The reference standards are the two radiologists' rating on plain radiography. Because the specificity of diagnosing mastoiditis is low on the plain radiography, the authors also provide the testing results based on radiologists' rating from CT images. The algorithms trained on 4139 patients and tested on three testing sets and showed the SENet trained on by multiple views images achieved better performance than that trained by single view images. Compare with reference standard from CT, the SENet showed similar sensitivity with radiologists and higher specificity than radiologists.

However, the authors need to find similar literature as benchmarks to compare the performance.

R1-1:Authors write all data are fully available without restriction. How about the image data and labels?

 All the mastoid series (images) were uploaded with labels (by both radiologists and algorithm). However, due to the very large volume of dataset (more than 9,000 image data sets) image data of training/validation set were compressed.

R1-2:The title is too general.

 According to your suggestion, we changed the title to “Performance of deep learning to detect mastoiditis using multiple conventional radiographs of mastoid”.

R1-3:Line 3 add anteroposterior (AP)

 Thank you for your kind comment. We added ”anterior-posterior”. (L3)

R1-4:Line 9 add case numbers for training and validation sets.

 According to your suggestion, we added case numbers.(L9)

R1-5:Line 15 add AUC values

 We added those values.(L16-19)

R1-6:Rephase the sentence on line 22.

 We appreciate your kind comment. We rephrased the sentence. “It could diagnose mastoiditis on mastoid series with similar to superior diagnostic performance to radiologists”“The diagnostic performance of the algorithm for detecting mastoiditis on mastoid series was similar to or higher than that of radiologists.” (L24-25)

R1-7:Line 43 This also contributes to a tremendous increase in radiologists’ workloads. What is this refer to?

 Thank you for your keen comment. It looks confusing. We changed the sentence to “Simple radiographies take a large portion of radiologists’ work-loads.” (L47-48)

R1-8:Line 66 What is the inclusion criteria in detail? Do somehow mastoiditis need a screening? Why the mastoid series didn't include any other diseases?

 Mastoid series were usually performed for screening mastoiditis (infection/inflammation) before operations such as organ transplantation and cochlear implantation. In those cases, preoperative treatment of mastoiditis is very important. (immunosuppresants will be used in patients after organ transplantation surgery, cochlear implant electrodes will go through mastoid air cells in cochlear implant op) Also, mastoid series were also performed in patients with suspected mastoiditis. (L69)

Mastoid series were used for detecting mastoiditis. Other diseases are very rare in mastoid air cells and also even those rare diseases (such as tumorous condition) are usually presented as mastoiditis patterns in imaging.

R1-9:What are the criteria for those patients underwent TB CT after radiography? Do you think those patients' plain radiographic images are different from the rest?

Actually majority of cases in gold standard test sets (which have both mastoid series and TB CT) performed TB CT and mastoid series simultaneously. Some clinicians performed both studies. (TB CT for precise diagnosis and mastoid series as base line study for following up because multiple/serial following up with CT is not good in terms of cost/effectiveness and ionizing radiation (As in pneumonia patients, who can perform chest CT at initial diagnosis and then serial following up studies are usually done using serial chest radiographies) We don’t think that those cases are different from others. Also we analyzed the data based on an individual ear (not per patient) and unilateral diseases are more frequent than bilateral diseases.Therefore, even if those cases had more mastoiditis than the others, contralateral normal mastoid air cells were also included in it.

R1-10:What if the two radiologists labeled different?

 Thank you for your keen comment. For those cases with disagreement, two radiologists discussed to reach a consensus. (L96)

R1-11:Show kappa coefficients between two radiologists for each dataset.

 According to your suggestion we inserted that information in Table 1.

R1-12:Why I.R. and L.S. instead of H.N.J. label the gold standard test set again or for CT instead of radiographic? How to get the final consensus?

 Because I.R. and H.N.J. labeled near 10,000 image sets (training and validation sets, external validation sets) during a relatively short period of time, we thought that there could be some kind of training effect and we wanted to check this out. (As a result, L.S. also showed similar diagnostic ability to I.R..) So we wanted to compare the ability of algorithm and other head and neck radiologist (L.S.) other than those two radiologists using gold standard test set.

It seems I.R. read both CT and plain radiography.Yes. However, labeling TB CT and labeling radiographs were performed separately (blinded). When labeling the plain radiographs of gold standard set by I.R., those plain radiographs (around 800 image sets) were randomly mixed with other 9,000 images in training/validation sets.

Because TB CT results are very straightforward, labeling TB CT results was done by one reader (I.R.).(L101)

R1-13:In the gold standard testing set, you made two types of labels, plain radiographic one and CT one. Please express clearly. And clarify it for each AUC result of gold standard testing set.

 We appreciate your insightful comment. We agree that we should clearly describe that.(L94-106)

We also added the information “based on labels by conventional radiography or standard reference (TB CT)” in Table 2, 3, 4.

R1-14:Line 102 Show a typical case for each category.

 According your suggestion, we inserted typical images. (figure 1)

R1-15:Line 122 Why did 120*180 mm resize to 384*256 pixels? How do you determine the center point? Is there any outliers?

Thank you for the comments. We corrected from120*180mm to 180*120mm. The left/right center points in the AP view are the points annotated in the image below. Outliers are excluded from the analysis.(L135)

R1-16:Line 126 Do horizontal and vertical shift on the original image or on the cropped image with zero paddings?

 Horizontal and vertical shift were used in the training process.

R1-17:Line 130 add CPU GPU types and memories.

 We added those types and memories.(L144-145)

R1-18: Draw the network architecture of single and multiple views, especially showing the difference of inputs.

For single view, learning rate decayed every 5000 steps at the rate of 0.94 with the initial value of 0.01. While those parameters change to 1000, 0.04, 0.005. Please explain the reason, or add new experiments to evaluate the parameter effects. The batch size is normal to set as 32,64,128etc instead of 12.

 According to your suggestion, we inserted the network architecture (figure 2). 

The weight of multiple view networks was fine-tuned using the weights obtained from the single view networks as initial values. Therefore, the initial value, decay rate, and decay step were reduced in this process. The batch size was set to 12 due to limitation of GPU memory. It was the maximum size that could be trained on one GPU. We added some explanations with relevant reference. (L162-170)

R1-19:The pediatric group is the most common. Age with mean and sd in table 1 are not good metrics. Maybe use percentage for each age group.

 Age is presented in groups instead of mean and sd. Also, number of each sex is corrected in Table 1. 

R1-20:Clarity what the Deep learning algorithm is in table 3, multi-views or single-view?

 Thank you for your keen comment. It is the result using multiple views. We added it.

R1-21:The authors use the cut-off point at which sensitivity was 95%, and the cut-off point at which specificity was 95%. However, in the table 3 and 4the metrics are not 95%.

 It means a cutoff point that was 95% sensitivity in the validation set, not in each test sets. Related texts and reference are added. (L236-238)

R1-22:Redesign table 4. It is hard to read. Maybe put sensitivity and specificity together for each point.

 According to your suggestion, we redesigned the table. The sensitivity and specificity values for each cutoff are presented.

R1-23:Which dataset is the images of Fig4 from? I think using gold-standard testing set is better due to the CT labels.

 We agree that it would be better to use images in the gold standard set. Therefore all the images except for Fig 4d (new Fig 6d) were from gold standard test set. However we could not find typical false negative images which have bilateral asymmetry on AP view in gold standard set. (No false negative images in gold standard set showed marked asymmetry on AP view)

R1-24:Line 372 typo "rained".Thank you for your kind comment. We corrected it. (L386)

R1-25:The authors did not show location results for "deep learning algorithm depicted the exact location of diseased mastoid air cells". Some selected images in Fig4 are not enough.

 Thank you for your comment. Only the images that the algorithm predicted as abnormal showed hot spots in class activation maps. Those spots corresponded to the mastoid air cells depicted in Fig 1.

According to your suggestion (R1-14), we inserted typical images of each category with annotation of the exact location of mastoid air cells (Fig 1)

R1-26:Line 375 rephase similar to superior... to

 According to your suggestion, we rephrased the sentence. “the diagnostic performance of the deep learning algorithm was similar to or higher than that of radiologists”.We also corrected another sentence in Conclusion section.(L378-379, L389-390)

Reviewer #2: The authors had presented a study to compare the diagnostic performance of deep learning algorithm trained by single view or multiple views. They evaluate the performance of the algorithms trained by the two strategies and also compared those trained algorithms with expert manual diagnostic performance. The conclusion of the study is that the deep learning algorithm trained by multiple views perform better than algorithms trained by single view. Also the algorithms trained by multiple views can achieve similar (or even better) performance than the radiologists.

R2-1:The conclusion drawn by the authors are meaningful, but the first part of the conclusion is predictable without the study. As stated by the authors, in practical, manual procedure would use multi-view instead of single view. Meanwhile, the accuracy of algorithms trained by using multi-view would be expected to outperform algorithms trained by using one single view. According to the ROC curves shown by the authors, even though the performances are significantly different according to statistical analysis , the accuracy numbers are not that different.

 We appreciate your insightful comments. We totally agree with you that we can easily expect that algorithms trained by multiple views outperform those using single view like humans do. However it is a kind of expectation and we don’t know exactly what the algorithms do if there is discrepancy between AP view and lateral views. So we just wanted to confirm that there is additive value of multiple views even though it looked like a somewhat obvious result.

R2-2:The authors have done thorough statistical analysis to support its claims. However, I believe the authors should clearly discuss the innovation or significance of this study. Currently, the paper gave out a signal that it proved an well-expected conclusion and there is limited to none innovation in the methodologies. This is a bit difficult to justify the significance of the work.

 Thank you for your keen comment.

Unlike previously studied (deep learning studies) simple radiographs (e.g., mammography of breast, waters views of paranasal sinuses, chest PA of lung), mastoid views are summations of many complex anatomic structures around mastoid air cells (e.g., temporo-mandibular joints, complex skull bases, and even auricles). Also mastoid air cells have considerable anatomic variations (even though breast, lung, sinuses have many variations, mastoid pneumatization patterns are much more diverse) between individuals and even in a person, they change greatly during age. Therefore, even radiologists are reluctant to interpret mastoid views if they are not trained head and neck radiologists. So we wanted to know whether the algorithm trained by data of around 5,000 patients can find regions of interest consistently (class activation map showed that) and even further whether it can predict those images well comparable to head and neck radiologists. The results showed its potential.Regarding that many parts of human bodies have huge anatomic variations, it is meaningful. And in terms of reducing radiologists’ work-loads, studies using simple radiographs (which have taken majority of medical images until now) are useful.

During performing this study, we also found that analyzing symmetry would be very important and helpful to improve the ability of algorithms (especially for very complex/diverse anatomic structures, Of course, this has been a frequently used method in interpreting medical images by radiologists). So we are doing related studies, now. This study is also a kind of bridging study to future studies. 

We added some of this content in Discussion.

R2-3:The paper is not well organized. Repetitive contents show up a lot. For instance, page 6, line 94-101 are repetitive. The paragraphs before Conclusion section are also poorly organized. The authors should re-organize the paper.

 The two paragraphs in “Labeling”(page 6) are not repetitive contents.

We agree that those paragraphs are quite confusing.

Firstly, training/validation set, temporal external test set, geographic external test set were labeled by two radiologists (I.R. and H.N.J). And gold standard test set was labeled based on TB CT results.

Then, to compare the diagnostic performance of algorithm with radiologists’ performance, two radiologists (I.R. and L.S.) labeled gold standard test set (mastoid series). However this time, another radiologist (L.S. instead of H.N.J) labeled gold standard set. Because I.R. and H.N.J. labeled near 10,000 image sets (training and validation sets, external test sets) during a relatively short period of time, we thought that there could be some kind of training effect and we wanted to check this out. (As a result, L.S. also showed similar diagnostic ability to I.R..)

This content was not described clearly in the manuscript and this made confusion. We rephrased some sentences.(L94-106)

We also reorganized the “Discussion” section (before Conclusion).

R2-4:One minor question: In page 8, why are the learning rate of the two CNN with very different decay rate (0.94 vs 0.04) and initial value (0.01 vs 0.005)?

 The weight of multiple view network was fine-tuned using the weights obtained from the single view networks as initial values. Therefore, the initial value and decay rate were reduced in the CNNs combined with multiple views. We added some explanations with relevant reference. (L162-170)

Reviewer #3: This manuscript explores mastoiditis classification with multiple view and single view and the comparison with radiologists. The manuscript is easy to understand and well-written. However, several limitations and points for further clarifications are listed below:

R3-1: Why using the patients w/ TB CT as the gold standard test set? Is there a diagnostic accuracy difference compare to multiple view? If yes, what’s the accuracy difference?

 The result of TB CT is very straight forward. If there’s soft tissue density in mastoid air cells and/or sclerotic change of mastoid bone, it is mastoiditis. However, mastoid series (plain radiography) are sometimes very difficult to tell mastoiditis from normal (as described in the manuscript, e.g., summation of complex anatomic structures, individual anatomic variations). Therefore two radiologists read the mastoid series together in all the other data sets. In the gold standard set, However, Cohen’s κ coefficients between TB CT label and radiologist 1 (I.R.) and TB CT label and radiologist 2 (L.S.) were substantial (both 0.66 (95% C.I.: 0.61-0.72)).

R3-2:The gold standard test set labeling is not clearly described in Page 6, line 94-101. Only until I read the result section, I start to realize how the labeling was conducted for the gold standard test set. I was confused by line 94 and line 100, as there are two types of labeling descriptions. Maybe the authors state ahead of that the gold standard test set was labeled twice, one was based on the concurrent TB CT by I.R. and H.N.J., the other time was based on mastoid series like in the training/validation set by I.R. and L.S.. Same for describing the labeling criteria for gold standard test set in Line 102-113, it is confusing that at the beginning saying “all the image in the dataset were labeled according…” and later on saying that “The gold standard test sets were labeled as … according to the results of TB CT.”

 Thank you for your detailed comments. We totally agree that it is quite confusing.

Firstly, training/validation set, temporal external test set, geographic external test set were labeled by two radiologists (I.R. and H.N.J). And gold standard test set was labeled based on TB CT results.

Then, to compare the diagnostic performance of algorithm with radiologists’ performance, two radiologists (I.R. and L.S.) labeled gold standard test set (mastoid series). However this time, another radiologist (L.S. instead of H.N.J) labeled gold standard set. Because I.R. and H.N.J. labeled near 10,000 image sets (training and validation sets, external validation sets) during a relatively short period of time, we thought that there could be some kind of training effect and we wanted to check this out. (As a result, L.S. also showed similar diagnostic ability to I.R..)

We described more explanation of this.(L94-106)

R3-3.How to control if the two neuroradiologists have different opinions on the same patient, and what if the labeling is different based on TB CT and mastoid series for the gold standard test set? Which one should be used as the final classification label?

 Two radiologists discussed the cases with different opinions to reach a consensus.(L96)

When comparing the abilities of algorithms using single view with those using multiple views, we used the labels by two radiologists (as in the other data sets (training/validation, temporal, geographic external test sets)). However, when comparing the ability of algorithm with radiologists’, we used the TB CT data as standard reference. (Because human (radiologists’) labeling should be also tested by standard reference in this comparison) We clearly described (added) this information in the table legends.(Tables 2-4)

R3-4.The deep learning method in Page 8 is not clearly described. How were the CNNs combined with multiple views? A structure figure is suggested for better illustration. Is that the CNN model is trained for each view respectively first, and further to average the last SE-ResNet module’s Log-Sum-Exp pooling values for all individual views to build the multiple view model? Is there a finetuning for the multiple view model? If yes, how did it conducted? If due to page/word count limitations, please include the details in a supplementary file.

 Thank you for your kind comments. We inserted the architectures of the multiple view model (Figure 2). The multiple view model was fine-tuned.The fine-tuning process for the CNNs combined with multiple views and related references were also added.(L154-170)

R3-5.In Table 1.the authors should give the full description of the abbreviations as notes. What does “CR” stands for? Please use a dash “-” to indicate the content is not available. The labels are different based on different imaging (CR and TB CT). Which is the final label of the gold standard test set, based on CR or CT? (refer to Question #3).

 Thank you for your kind recommendations. CR stands for conventional radiography. We also added dashes in the blanks. (Table 1)

(same as the answer to R3-3) When we compared the abilities of algorithms using single view with those using multiple views, we used the labels by two radiologists (as in the other data sets). However, when we compared the ability of algorithm with radiologists’, we used the TB CT data as standard reference. (Because human (radiologists’) labeling should be also tested by standard reference in this comparison) We clearly described (added) this information in the table legends.(Tables 2-4)

R3-6.The notations of Figure 4 are not clear, I’m assuming the left side is the input image, and the right side is the outputs based on the attentions. It will be much clear if the authors can circle/point out where the lesions are in true positive (a), false negative (d), and postoperative state (e).

 We appreciate your comments. Right side images are original images and left side images are class activation maps. Only the cases that the algorithm predicted as abnormal showed hot spots on the class activation maps. (Nothing was detected on class activation maps in cases classified as normal by the algorithm.) We thought that it would be better to annotate the location of mastoid air cells on AP view and lateral view separately than to overlay the annotations on the Fig 4 (new Fig 6). So we added (new) typical images of each category with annotations (red or white circles) of the exact locations of mastoid air cells in Fig 1.

R3-7.It’s suggested to provide the confusion matrix like in Fig.2 but for based on the mastoid series.

 Confusion matrices between the conventional radiography (mastoid series) based label and predictions of the deep learning algorithm are shown in Fig 5 (previous Fig 3).

R3-8.A normal/abnormal case is based on an individual patient or an individual ear? Is diagnostic accuracy calculated as ear-based or patient-based? If one patient has both ears as otomastoiditis, how can the authors determine the classification accuracy if the results show one ear is positive and another ear is negative?

 All the data including diagnostic performance are based on an individual ear (not per patient)

---

## [Decision Letter · Decision Letter 1]

9 Sep 2020

PONE-D-20-11571R1

Performance of deep learning to detect mastoiditis using multiple conventional radiographs of mastoid

PLOS ONE

Dear Dr. Ryoo,

Thank you for submitting your manuscript to PLOS ONE. After careful consideration, we feel that it has merit but does not fully meet PLOS ONE’s publication criteria as it currently stands. Therefore, we invite you to submit a revised version of the manuscript that addresses the points raised during the review process.

We look forward to receiving your revised manuscript.

Kind regards,

Yuchen Qiu, Ph.D.

Academic Editor

PLOS ONE

Reviewers' comments:

Reviewer's Responses to Questions

**Comments to the Author**

1. If the authors have adequately addressed your comments raised in a previous round of review and you feel that this manuscript is now acceptable for publication, you may indicate that here to bypass the “Comments to the Author” section, enter your conflict of interest statement in the “Confidential to Editor” section, and submit your "Accept" recommendation.

Reviewer #1: (No Response)

Reviewer #3: All comments have been addressed

2. Is the manuscript technically sound, and do the data support the conclusions?

Reviewer #1: Yes

Reviewer #3: Yes

3. Has the statistical analysis been performed appropriately and rigorously? 

Reviewer #1: Yes

Reviewer #3: Yes

4. Have the authors made all data underlying the findings in their manuscript fully available?

Reviewer #1: Yes

Reviewer #3: Yes

5. Is the manuscript presented in an intelligible fashion and written in standard English?

Reviewer #1: Yes

Reviewer #3: Yes

6. Review Comments to the Author

Reviewer #1: The author partially address my concerns. The revision on the main text should be as important as comment response.

The authors claim: “because the most commonly affected age group is the pediatric group, especially patients under two years old who are very sensitive to radiation exposure, simple radiography still has its role.”

But the age distribution in table 1 is not the case. 50-59 year-old patients contribute the most. Less than 20 year-old patients is minor, only around 8%. This compromised the novelty. Please explain it.

“Deep learning algorithm depicted the exact location of diseased mastoid air cells.” The activation map of diseased mastoid air cells is a key contribution of this article. However, this article lacks of the detail implement of activation map in the method. Please add it.

Please provide a reference for the claim of “Simple radiographies take a large portion of radiologists’ work-loads.” (L47-48)

R1-8:Line 66 What is the inclusion criteria in detail? Do somehow mastoiditis need a screening? Why the mastoid series didn't include any other diseases?

 Mastoid series were usually performed for screening mastoiditis (infection/inflammation) before operations such as organ transplantation and cochlear implantation. In those cases, preoperative treatment of mastoiditis is very important. (immunosuppresants will be used in patients after organ transplantation surgery, cochlear implant electrodes will go through mastoid air cells in cochlear implant op) Also, mastoid series were also performed in patients with suspected mastoiditis. (L69)

Mastoid series were used for detecting mastoiditis. Other diseases are very rare in mastoid air cells and also even those rare diseases (such as tumorous condition) are usually presented as mastoiditis patterns in imaging.

Please add this background information to the paper.

Actually majority of cases in gold standard test sets (which have both mastoid series and TB CT) performed TB CT and mastoid series simultaneously. 

Please add this response to the paper.

L96 please revise to labels were determined by consensus after the two radiologists discussed.

It seems I.R. read both CT and plain radiography.Yes. However, labeling TB CT and labeling radiographs were performed separately (blinded). When labeling the plain radiographs of gold standard set by I.R., those plain radiographs (around 800 image sets) were randomly mixed with other 9,000 images in training/validation sets.

Please add this response to the paper.

The left/right center points in the AP view are the points annotated in the image below. Outliers are excluded from the analysis.(L135)

Please add this response to the paper.

R1-16:Line 126 Do horizontal and vertical shift on the original image or on the cropped image with zero paddings?

 Horizontal and vertical shift were used in the training process.

The authors didn’t answer this question.

Image shift will result in black edge. Shifting on the original image before cropping will replace the black edge with the pixels adjacent to the edge.

Did you do it on the original image or on the cropped image?

Reviewer #3: (No Response)

7. PLOS authors have the option to publish the peer review history of their article (what does this mean?). If published, this will include your full peer review and any attached files.

Reviewer #1: **Yes: **Jingchen Ma

Reviewer #3: No

---

## [Author Response · Author response to Decision Letter 1]

6 Oct 2020

Reviewer #1: The author partially address my concerns. The revision on the main text should be as important as comment response.

R1-1:The authors claim: “because the most commonly affected age group is the pediatric group, especially patients under two years old who are very sensitive to radiation exposure, simple radiography still has its role.”

But the age distribution in table 1 is not the case. 50-59 year-old patients contribute the most. Less than 20 year-old patients is minor, only around 8%. This compromised the novelty. Please explain it.

 Thank you for your keen comment. It is known that the pediatric group can be more commonly affected by otomastoiditis than the older age groups (Ref. 2&6). However, the “affected” age here does not necessarily mean the most prevalent age group in radiologic examinations. Relative data shortage in this group may arise from other reasons. Unlike adult patients, many clinicians usually treat young patients without radiologic examinations in the first place. This is probably because young patients are more sensitive to radiation exposure than adults (Even though the radiation exposure of simple radiographs is much less than that of CT examinations) and/or performing examinations for children often involves various technical difficulties. Also, because organ transplantation surgeries are usually performed to adult patients, preoperative work up is much less frequent in children. 

Even though our network was trained with the data set containing less exams of young patients, the evaluation result on the age group under 20 (accuracy=0.888, AUC=0.918) shows statistically non-inferior performance (p=0.928, p=0.125) compared with the other age group (accuracy=0.864, AUC=0.958); thus, the most affected group to the otomastoiditis can sufficiently benefit from the developed algorithm as well.

R1-2 “Deep learning algorithm depicted the exact location of diseased mastoid air cells.” The activation map of diseased mastoid air cells is a key contribution of this article. However, this article lacks of the detail implement of activation map in the method. Please add it.

We appreciate your comment. The process of obtaining the class activation mapping in the deep learning algorithm part was described in more detail. In the manuscript, the class activation mapping obtained from CNN for multiple views was presented, and related parts were added. (L187-195, L322) In addition, the code we implemented was uploaded to the public repository and the related part was added. (L184-185)

R1-3 Please provide a reference for the claim of “Simple radiographies take a large portion of radiologists’ work-loads.” (L47-48)

As we mentioned in the manuscript, simple radiographs are the most commonly performed imaging procedures in the US and the average radiologist reads more than 100 chest radiographs per day. (Ref. 7,8) However those references focused on chest radiographies (of course, the chest radiographs take major part of all radiographs). So we added another reference (Ref 9: Trends in Diagnostic Imaging Utilization among Medicare and Commercially Insured Adults from 2003 through 2016, Radiology 2020:294:342-350) which showed that more than a half of the diagnostic imaging performed from 2003 through 2016 in the US are simple radiographies. In this article, the diagnostic imaging modalities included even non-radiologists’ works (e.g., echocardiography, Nuclear imaging). Nonetheless, more than a half were simple radiographies. (L48)

R1-4:Line 66 What is the inclusion criteria in detail? Do somehow mastoiditis need a screening? Why the mastoid series didn't include any other diseases?

 Mastoid series were usually performed for screening mastoiditis (infection/inflammation) before operations such as organ transplantation and cochlear implantation. In those cases, preoperative treatment of mastoiditis is very important. (immunosuppresants will be used in patients after organ transplantation surgery, cochlear implant electrodes will go through mastoid air cells in cochlear implant op) Also, mastoid series were also performed in patients with suspected mastoiditis. (L69)

Mastoid series were used for detecting mastoiditis. Other diseases are very rare in mastoid air cells and also even those rare diseases (such as tumorous condition) are usually presented as mastoiditis patterns in imaging.

Please add this background information to the paper.

 Thank you for your kind comment. We added this content in the manuscript. However, some of the contents are too long and a little bit out of focus in M&M section (e.g., “other diseases such as primary tumors are very rare in the mastoid area and even those diseases are presented as mastoiditis” is a common sense to doctors), we added some parts of these contents. (L73-76)

R1-5 Actually majority of cases in gold standard test sets (which have both mastoid series and TB CT) performed TB CT and mastoid series simultaneously. 

Please add this response to the paper.

 According to your suggestion, we added this information in the manuscript. (L81-82)

R1-6 L96 please revise to labels were determined by consensus after the two radiologists discussed.

 We appreciate your kind suggestion. According to this suggestion, we revised the sentence. (L103-104)

R1-7 It seems I.R. read both CT and plain radiography. Yes. However, labeling TB CT and labeling radiographs were performed separately (blinded). When labeling the plain radiographs of gold standard set by I.R., those plain radiographs (around 800 image sets) were randomly mixed with other 9,000 images in training/validation sets.

Please add this response to the paper.

 Thank you for the comments, we added this response in the manuscript. (L114-116)

R1-8 The left/right center points in the AP view are the points annotated in the image below. Outliers are excluded from the analysis.(L135)

Please add this response to the paper.

 We added this content in the manuscript.(L147-149) The AP view image of the points annotated was also added (Fig 2).

R1-9:Line 126 Do horizontal and vertical shift on the original image or on the cropped image with zero paddings?

 Horizontal and vertical shift were used in the training process.

The authors didn’t answer this question.

Image shift will result in black edge. Shifting on the original image before cropping will replace the black edge with the pixels adjacent to the edge.

Did you do it on the original image or on the cropped image?

 We certainly did the shift operation on the “original” images and avoided the black edge problem. Thank you for reminding us.

---

## [Decision Letter · Decision Letter 2]

21 Oct 2020

Performance of deep learning to detect mastoiditis using multiple conventional radiographs of mastoid

PONE-D-20-11571R2

Dear Dr. Ryoo,

We’re pleased to inform you that your manuscript has been judged scientifically suitable for publication and will be formally accepted for publication once it meets all outstanding technical requirements.

Kind regards,

Yuchen Qiu, Ph.D.

Academic Editor

PLOS ONE

Additional Editor Comments (optional):

Reviewers' comments:

Reviewer's Responses to Questions

**Comments to the Author**

1. If the authors have adequately addressed your comments raised in a previous round of review and you feel that this manuscript is now acceptable for publication, you may indicate that here to bypass the “Comments to the Author” section, enter your conflict of interest statement in the “Confidential to Editor” section, and submit your "Accept" recommendation.

Reviewer #1: All comments have been addressed

2. Is the manuscript technically sound, and do the data support the conclusions?

Reviewer #1: Yes

3. Has the statistical analysis been performed appropriately and rigorously? 

Reviewer #1: Yes

4. Have the authors made all data underlying the findings in their manuscript fully available?

Reviewer #1: Yes

5. Is the manuscript presented in an intelligible fashion and written in standard English?

Reviewer #1: Yes

6. Review Comments to the Author

Reviewer #1: The authors have addressed my comments. Now the paper is in good shape. I suggest to accept this paper.

7. PLOS authors have the option to publish the peer review history of their article (what does this mean?). If published, this will include your full peer review and any attached files.

Reviewer #1: No

---

## [Editor Report · Acceptance letter]

23 Oct 2020

PONE-D-20-11571R2 

Performance of deep learning to detect mastoiditisusing multiple conventional radiographs of mastoid 

Dear Dr. Ryoo:

I'm pleased to inform you that your manuscript has been deemed suitable for publication in PLOS ONE. Congratulations! Your manuscript is now with our production department. 

Kind regards, 

on behalf of

Dr. Yuchen Qiu 

Academic Editor

PLOS ONE